# Deep Learning Models to Predict Finishing Pig Weight Using Point Clouds

**DOI:** 10.3390/ani14010031

**Published:** 2023-12-21

**Authors:** Shiva Paudel, Rafael Vieira de Sousa, Sudhendu Raj Sharma, Tami Brown-Brandl

**Affiliations:** 1Department of Biological Systems Engineering, University of Nebraska-Lincoln, Lincoln, NE 68583-0726, USA; spaudel6@huskers.unl.edu (S.P.); raj.sharma@unl.edu (S.R.S.); 2Department of Biosystems Engineering, University of Sao Paulo, Pirassununga 13635-900, SP, Brazil; rafael.sousa@usp.br

**Keywords:** 3D deep learning, PointNet, weight estimation

## Abstract

**Simple Summary:**

Monitoring the weight of farm pigs is crucial for their well-being. Implementing Cameras and Machine Vision Systems shows promise for automating this process. Traditionally, studies have focused on RGB and depth images for weight prediction, using measurements like volume and body area. However, these methods prove less robust in fluctuating environmental conditions, especially lighting. This study reveals that PointNet, a 3D deep learning architecture trained on point cloud data (3D points), outperforms the conventional approach, and demonstrates stability in varying light conditions due to its ability to learn on spatial information. This finding underscores the potential for PointNet to significantly improve the accuracy and reliability of weight monitoring in farm settings.

**Abstract:**

The selection of animals to be marketed is largely completed by their visual assessment, solely relying on the skill level of the animal caretaker. Real-time monitoring of the weight of farm animals would provide important information for not only marketing, but also for the assessment of health and well-being issues. The objective of this study was to develop and evaluate a method based on 3D Convolutional Neural Network to predict weight from point clouds. Intel Real Sense D435 stereo depth camera placed at 2.7 m height was used to capture the 3D videos of a single finishing pig freely walking in a holding pen ranging in weight between 20–120 kg. The animal weight and 3D videos were collected from 249 Landrace × Large White pigs in farm facilities of the FZEA-USP (Faculty of Animal Science and Food Engineering, University of Sao Paulo) between 5 August and 9 November 2021. Point clouds were manually extracted from the recorded 3D video and applied for modeling. A total of 1186 point clouds were used for model training and validating using PointNet framework in Python with a 9:1 split and 112 randomly selected point clouds were reserved for testing. The volume between the body surface points and a constant plane resembling the ground was calculated and correlated with weight to make a comparison with results from the PointNet method. The coefficient of determination (R^2^ = 0.94) was achieved with PointNet regression model on test point clouds compared to the coefficient of determination (R^2^ = 0.76) achieved from the volume of the same animal. The validation RMSE of the model was 6.79 kg with a test RMSE of 6.88 kg. Further, to analyze model performance based on weight range the pigs were divided into three different weight ranges: below 55 kg, between 55 and 90 kg, and above 90 kg. For different weight groups, pigs weighing below 55 kg were best predicted with the model. The results clearly showed that 3D deep learning on point sets has a good potential for accurate weight prediction even with a limited training dataset. Therefore, this study confirms the usability of 3D deep learning on point sets for farm animals’ weight prediction, while a larger data set needs to be used to ensure the most accurate predictions.

## 1. Introduction

Live weight measurement is an important factor in the management of farm pigs [1,2]. Body weight provides important information regarding the animal’s health and well-being [3]. It is noteworthy that in commercial production environments, obtaining accurate records of individual animal weights often encounters challenges [4]. Nevertheless, the systematic capture of individual weights plays a pivotal role in decisions pertaining to swine marketing [5]. Furthermore, continuous weight monitoring can provide the opportunity to promptly identify individuals potentially experiencing digestive and health issues [6]. 

Automatic weight prediction based on computer vision has been investigated for predicting animal weight in production systems in a non-invasive and economical way [7,8]. Digital image processing is the most widely used vision-based weight prediction method. In digital image processing, images of pigs’ bodies are analyzed, and animal dimensions such as area, body length, width, and chest depth are calculated and correlated to animals’ weight [9,10,11,12,13]. In the mid-nineties, Brandl et al. [14] analyzed pigs’ body areas from digital images to estimate their weights, and were able to predict with less than 6% deviation [14]. Although digital image processing has shown great results, it comes with challenges, such as pigs needing to be under the camera at a somewhat predetermined position, color patterns or dirt on the animals, and lighting conditions needing to be consistent [6,15]. 

To overcome the challenge faced by digital image analysis, depth images have been used to extract features of pigs’ bodies [16,17]. Depth images extract three-dimensional (3D) features of an animal’s body such as height, volume, and curvature [18]. Pezzulo et al. [19] used Microsoft Kinect to measure heart girth, length, and height, and used linear and nonlinear models to get the coefficient of determination (R^2^ > 0.95). Manual feature extraction is insufficient for a precise prediction as a limited number of parameters can be extracted and a small change in posture alters all of the parameters [17]. Okayama et al. [17] tried extracting information about spinal cord bends to make their model stable even with the change in posture. Although their work has shown an improvement in prediction, this approach is strictly restricted to implementation in the predefined close space. Deep learning approaches are presented as a solution to make vision-based weight estimation reliable even with slight changes in the postures of animals.

Artificial Neural Network (ANN) and deep learning have been implemented on digital RGB images of animals to estimate their weights [15,20]. Hidden fully connected or 2D convolution layers have been used to extract the features from the image with an additional regression layer at the end to estimate weight. Jun et al. [21] tried to address the problem of posture and lighting by introducing curvature and deviation, but they could only achieve a coefficient of determination of (R^2^ < 0.79). In some approaches, depth images were colorized based on the height of the animal before presenting it to ANN [22]. The benefit from the colorization is very minimal and becomes similar to that of using RGB images. 

Furthermore, all these approaches are common in using predefined lab setups to collect pigs’ RGB or depth images. While studies have shown a correlation up to (R^2^ = 0.99) with RSME below 1 kg, their constrained experimental design makes their implementation in natural farms impossible [16,22]. There is thus a strong need for a method that can effectively predict animal weight in natural farm settings and conditions. In recent days 3D deep learning methods have been studied for precision livestock farming such as identification [23], posture classification [24], and weight prediction [25,26] as they are robust even with changes in posture and lighting conditions. However, these methods have yet to be tested in commercial farms.

The objectives of this study were to: (1) develop and implement a 3D deep learning method to predict individual pig weight from the point cloud collected with free movement as a typical pen environment without controlled illumination; (2) compare the use of deep learning modeling with a most-studied method of weight prediction using machine vision (in other words, linear regression model between pig weight and volume); and (3) analyze how well these methods perform for different weight groups. 

## 2. Materials and Methods

The animal care and handling techniques were approved by the Animal Use Ethics Committee of the Faculty of Animal Science and Food Engineering at the University of São Paulo (FZEA-USP), under protocol 6526160720. The animals were housed in covered outdoor pens; therefore the lighting was not controlled and varied throughout the day (Figure 1). These natural lighting conditions mimic some commercial conditions in certain parts of the world.

A total of 249 unrestrained grow-finish pigs (20–120 kg) were used in an experiment to estimate body weight. This experiment included walking pigs through an individual holding pen and capturing point clouds of these animals. The data was used to estimate the weights of these individual animals. Details of the experiment including the experimental setup, data collection, extraction, and preprocessing of 3D images (point clouds) are included below. In addition, specific methodology employed for volume calculation and the implementation of PointNet are described.

### 2.1. Data Collection

The collection of 3D videos and weighing of pigs was carried out in experimental farms at FZEA-USP. A total of 249 grow-finish pigs (7 to 20 weeks old) were housed in ten pens, with 25 animals per pen (20–120 kg). The animals had ad libitum access to food and water. On an individual basis, the weights of pigs were collected and 3D videos were capture on five different dates (5 August, 28 October, 4 November, and 9 November). 

The 3D video data acquisition was completed using a portable station composed of a 3D camera (Intel Real Sense D435 Stereo Depth Camera© (Intel Corporation, Santa Clara, CA, USA) and a computer. These top-down videos were recorded when pigs were freely moving in a holding pen after the weight was taken. A total of 249 videos of individual pigs were collected (each with a duration of 3–5 s), with the intention of extracting seven point clouds of each animal. 

### 2.2. Experimental Setup

Figure 1 provides a detailed depiction of the experimental setup, focusing on its side view. The camera used in the experiment was positioned at a height of 2.7 m from the floor, and was connected to a computer positioned laterally in the experimental setup via a USB 3.0 cable with a 5 m active extender. Manual triggering was employed for video acquisition utilizing Intel^®^ RealSense™ Viewer software version 2.47.

Camera parameter calibration was autonomously performed on-site using the Intel^®^ Depth Quality Tool software v2.50.0, incorporating plane fit, on-chip self-calibration, and tare calibration methodologies. Video captures of each specimen were conducted at 30 frames per second (fps), employing auto-exposure mode for both the color image sensor (color sensor) and the infrared sensor (depth sensor). The field of view for the color image was 69.4 × 42.5 (H × V) with a resolution of 640 × 480, while the field of view for the depth image was 86 × 57 (H × V) with a resolution of 848 × 480.

The holding pen in which the pigs were observed boasted dimensions with a major axis extending to 2.9 m and a minor axis measuring 1.5 m. Throughout the data collection period, each individual pig was allowed unrestricted movement within the holding pen. The camera captured their dynamic motions within this confined space. The varying positions and standing posture of the pigs within the pen influenced the visibility of different parts of their bodies to the camera as showcased in Figure 1b–e, creating variability in the projected volume of the animal.

### 2.3. Image Extraction and Preprocessing

The first step of data processing consisted of extracting 3D images (point clouds) from the collected 3D videos. For this, the Intel Realsense Viewer© (Intel Corporation, Santa Clara, CA, USA) application was used (the same application that was used to collect the 3D videos). At this stage, collected 3D videos were locally stored, and five to ten 3D images were manually extracted. The number of extracted images was dependent on the quality of the video.

The entire set of filters and functions developed allowed the 3D images to be read, aligned, filtered, converted, and stored (Figure 2). Points whose values on the *z*-axis were below or above pre-defined limits were removed according to the minimum and maximum average depth found in the 3D images of the pigs. In this way, the points referring to the floor and the side walls were removed. Points referring to structures that are outside the area of interest (that is, any point that is not referring to the swine) were also extracted through “xy” points that were outside of a pre-defined polygon. Further extraction was performed by removing points whose color was different and did not belong to the swine. The color threshold was found automatically by the Otsu method. It should also be noted that the collected data also included point clouds with some unknown noise pattern as demonstrated in Figure 3.

Finally, the point clouds were randomly subsampled to 1500 points, and removal of outliers took place to obtain the final extracted point cloud of the pig. The weights of the experimented pigs ranged from 20–120 kg. The range was further divided into three different groups: below 55 kg, 55–90 kg, and above 90 kg, in order to analyze whether weight has an effect on correlation as observed in other manuscripts [14]. Correlations were then calculated within each weight range to determine which range could best be explained by volume. The processed data is composed of a total of 1298 point clouds, out of which 644 weigh below 55 kg, 322 are between 55 and 90 kg, and 332 are above 90 kg. 

### 2.4. Volume Calculation

The literature has shown that the volume of the pig’s body directly correlates to its mass [16]. The volume was calculated as a projected volume between the pig’s top view (Figure 2d) and the pen floor, considered a constant 2.7 m from the camera. The volume was calculated using CloudCompare v2.11.3 software’s 2.5D volume function. The grid with step size 0.01 was projected in the ‘Z’ direction. The maximum height on the body from constant plane ‘L’ was considered as the height of the point when multiple points fall inside a cell. 

The pigs in the study had the freedom to move in any direction within the holding area, including moving their heads up and down. When the pigs lowered their heads, it resulted in limited or no contribution to the overall volume of the pig’s body. Conversely, a flat head posture significantly increased the volume, and different ear postures also caused changes in volume. To address the variance in volume caused by head movement, the head of the pigs was manually removed by cropping out any portion above the neck region in all of the point clouds. The postural variations depicted in Figure 1 posed challenges for automating head and neck removal. To ensure consistency across all selected images, a visual inspection was conducted to identify the narrowest width of the neck area. Subsequently, any portions above this defined narrow width were cropped out. 

### 2.5. PointNet

PointNet is a recently proposed algorithm that directly takes points for training [27,28]. PointNet has shown great potential in 3D classification and segmentation [29,30], but it has never been explored for livestock weight prediction. A significant advantage is that it directly takes a set of points as input and extracts features of the point clouds. The PointNet architecture takes a point cloud with *n* points as input and utilizes input transformation to extract features (Figure 4). The max pooling layer was used to aggregate the point features and finally K-score classes for classification. The input transformation layer is a mini-PointNet network that transforms point clouds into another coordinate system with the same dimension. T-Net is specifically designed to address the challenge of transforming the input point cloud to a canonical or normalized form. It achieves this by learning an optimal linear transformation matrix. This matrix is applied to the local features obtained from the shared MLP, effectively aligning and normalizing the point cloud. Learning and applying these transformations makes the network more robust, ensuring that it can recognize and understand 3D shapes regardless of their orientation or position in space.

Successive multilayered perceptron layers (MLP) were shared by each point on the cloud. The max pooling layer was used to aggregate the global point features vectors. Finally, the global point features were then passed through another MLP with a regression layer on top to output regress weight. The feature transformation layer in between successive shared MLP was removed for computational simplicity. The top layer for K-score classification was replaced with a fully connected dense layer with a RELU activation function that outputs weight. After preprocessing and selection of the point clouds, the PointNet architecture was trained. The PointNet algorithm was implemented on Google Colab GPU Python version 3.7.15 platform with Keras package (Version 2.9.0) using TensorFlow (version 2.9.2) as the backend. Open3d (version 0.16.0) package was used to visualize and read point clouds corresponding to weight levels. PointNet implementation on TensorFlow Keras by David Griffiths was taken as a reference for algorithm implementation. Adam optimizer with a learning rate of 0.01 was used. Root Mean Square Error (RMSE) was used as accuracy metrics and Mean Square Error (MSE) was used as the loss function. 

The model was trained for 1000 epochs with early stopping callback with patience of 10. In order to create more variation in the training dataset, data augmentation was used. The points were jittered between −0.005 and +0.005 to introduce variability in the training data while keeping the labels the same. The dataset was split into 9:1 for training and validation. A testing dataset containing 112 different point clouds was randomly selected from the same 249 pigs and was not used for training or validation. The performance of the models was evaluated by comparing the measured values of the target attributes with their respective predicted values based on the determination coefficient (R^2^) and RMSE. The iterative fine-tuning process employed in the training phase of the models allowed for the definition of optimal values for the hyperparameters of each supervised learning algorithm. This training and evaluation method ensured the generation of models with the ability to generalize to data not belonging to the database used for training the models.

## 3. Results and Discussion

A total of 1298 point clouds from 249 pigs were collected for this study. Of these, 644 corresponded to weights less than 55 kg, 323 ranged between 55 and 90 kg, and 333 were above 90 kg. A total of 112 point clouds were manually selected for volume correction. To achieve the highest possible correlation between the volume of animals and the weight-adjusted volume (calculated by removing the head and neck), an additional 112 point clouds from the total of 1298 were reserved specifically for PointNet testing. The remaining 1186 point clouds were then divided into a 9:1 ratio for training and validation. The predicted weight of animals from PointNet was subsequently correlated with the scale weight. The best prediction was achieved with Pointnet, exhibiting a coefficient of determination of 0.94. Following this, the results were further investigated by segmenting the point cloud into three different weight groups.

### 3.1. Volume to Weight Correlation

Given the quality of data, a subset of point cloud was created for volume correlation. In order to select point clouds for volume correlation, all 1298 point clouds were checked manually and those with minimal quality issues were selected. Out of 246 pigs, 112 best-point clouds, which do not have wavy patterns or missing big chunks of points (Figure 3), were selected for volume correlation. The selected pigs had a uniform weight distribution in the 20–120 kg weight range. 

Initially, the correlation between volume and weight was investigated. It was found that an R^2^ value of 0.74 was obtained when the complete volume of the pig (including the head) was correlated to the weight (Figure 5). To minimize the effect of free movement, the head of the pigs was removed and volume was calculated. The correlation improved to R^2^ = 0.76, as shown in Figure 6. This correlation did not come close to that which was reported by Condotta et al. [16] (R^2^ = 0.99) and Brandl and Jørgensen [14] (R^2^ = 0.98) in their experiment. It is evident that the point clouds under investigation were taken at different positions and standing postures (Figure 1), with some taken directly under the camera and some towards the edge of the holding barn, which led to variation in the volume of pigs compared to the stationary experiment approach followed in most of the literature. Although the weight seemed to be varying with volume, the changes in the volume itself with the movement of pigs within the barn led to less efficient prediction compared to the previously mentioned studies. 

Looking at different weight classes, the weight class below 55 kg was found to be best explained by the volume, with R^2^ = 0.77, whereas for higher weight classes the correlation was found to be decaying. The correlation between weight and volume for the 55–90 kg weight class was found to be R^2^ = 0.26, and for weight class above 90 kg, the weight could not be explained by pigs’ body volume as the correlation was only R^2^ = 0.01. 

One likely reason for this declining correlation could be attributed to errors occurring during the measurement of scale weight. The larger the animals, the more challenging it was to contain their motion within the scale for accurate weight measurement. Another potential factor contributing to the declining results could be the distribution of data. In weight classes below 55 kg, the average weight of animals was 41.17 kg. For the weight range of 55–90 kg, the average weight of animals was 67.74 kg, and for weight classes above 90 kg, the average weight of animals was 99.31 kg. This indicates that a higher number of animals were concentrated towards the lower range of classes 55–90 kg and above 90 kg, leading to a higher prediction error.

### 3.2. PointNet for Estimating Weight

One hundred and twelve point clouds were set aside for testing, and the remaining were split into 9:1 for training and validation. Changes in RMSE was recorded throughout the training period, and Figure 7 shows the learning curve of the change in RSME. The minimum RMSE of 6.03 kg was obtained at the 28th epoch of the training. Figure 8 shows point clouds along with true labels and predicted labels. Some of the point clouds as shown in Figure 3 were missing large quantities of points from the body area because of some unknown problem in the experimental setup. Those point clouds were considered good and included in training and testing. While point clouds with prominent noise pattern were not included for the volume correlation, the data set still might have smaller noise leading to poor correlation; however, the deep learning model is strong enough to predict the weight even with such points. 

One hundred and twelve random different point clouds were used for the testing. The trained model was presented with the reserved test data for prediction. Figure 9 shows correlation between scale weight and the predicted weight. It achieved an overall R^2^ of 0.94 with an RMSE of 6.87 kg. However, although the prediction was better compared to the volume correlation, the performance could have been better with bigger datasets. 

For the weight group below 55 kg, the model achieved an RMSE of 4.50 kg and an R^2^ of 0.81. The performance of predictions for the weight class 55–90 kg had an RMSE of 8.20 kg and an R^2^ of 0.71. For the weight class above 90 kg, the model achieved an R^2^ of 0.44 and an RMSE of 9.95 kg. The trend was similar to the volume correlation, where the correlation worsened with an increase in weight. As mentioned, the training dataset for the below 55 kg class was almost twice as large as the other classes, which likely resulted in significantly better performance. However, for the two remaining classes with a similar number of training data, the lower weight class (between 55–90 kg) performed better than the above 90 kg weight class. This indicates that even with PointNet, the prediction declined with larger pigs.

### 3.3. Comparison between Deep Learning and Volume-Based Regreesion

The PointNet architecture outperformed volume correlation in each class. Table 1 presents the results for all three classes along with overall correlation. It was observed that weight prediction becomes more difficult as the pigs grow larger. It was observed that difficulties in handling animals (constraining their movement during weight measurement leading to errors on ground truth data) might have contributed to higher errors on larger animals. Along with the difficulties in weight measurement, the low accuracy of prediction on large animals can be tied to the nonlinearity of the relationship between the weight and volume of large pigs as suggested by [16].

As discussed in the Section 2, the variability in position within the pen area and variability in standing posture might have been the major cause of poor performance of weight prediction using volume. This variability in positioning affected the projected volume of the pigs’ bodies, as highlighted in Figure 1c–e. Looking at Figure 1c, in this instance, one side of the pig’s surface was more prominently visible than the other. This observation emphasizes the influence of the pig’s spatial orientation within the pen on the visual representation of its anatomy.

The positioning within the pen area is a crucial factor affecting the projected volume of the pig. This is a critical component of this experiment, shedding light on the dynamic nature of their appearances as captured by the camera. 

Furthermore, beyond the impact of body visibility to the camera, the variation in the standing posture of the pigs introduces an additional layer of nonuniformity to the perceived volume. Figure 1c–e distinctly illustrates three distinct standing body postures. Notably, these postural variations have a direct and discernible effect on the volume captured by the depth sensor [31].

Examining Figure 1b–d, it becomes evident that changes in posture significantly influence the volume discernible to the depth sensor. The dynamic nature of these postural adjustments presents a challenge when employing computer vision for weight estimation, particularly in scenarios where the pigs are not constrained. The complexity introduced by the interplay of body visibility and postural changes underscores the intricacies involved in accurately estimating the weight of freely moving pigs through computer vision methodologies. This complexity might have led to most experimental setup designs where pigs are confined within a small area. Additionally, it should be noted that in the experimental setup resembling a commercial barn with numerous non-constrained pigs, segmenting individual animals, and calculating the volume of the animal becomes difficult, resulting in inaccurate predictions.

The PointNet approach employed in this study has proven to be effective in addressing the challenges encountered by volume correlation methods. Unlike traditional approaches, PointNet excels in learning features directly from raw, unordered point cloud data. This unique capability enables the model to capture intricate patterns and relationships within the data without the need for predefined structures or order making this method more robust to variation in animals’ body projection to the camera.

The PointNet predictions for the worst-performing weight class (above 90 kg) had an RMSE of 9.95 kg. In contrast, the volume-based prediction had a similar RMSE of 9.16 kg in the best-performing weight class (below 55 kg). The results clearly demonstrate that implementing PointNet and other 3D deep learning architectures lead to more accurate weight predictions. Furthermore, it should be noted that the PointNet architecture was trained with only 1186 point clouds, each consisting of only 1500 body surface points. Adding more point clouds for training will certainly improve the prediction accuracy.

Furthermore, it is important to note that the data for this study was collected with only one pig inside the holding pen. However, in farm settings, many pigs are typically housed together in a single pen. Therefore, further studies should be designed to capture and predict the weights of multiple pigs simultaneously. This can be achieved by placing a camera over the drinker or feeder of the pens and segmenting individual pigs in each frame. This approach will allow the algorithm to predict the weights of multiple pigs at once.

The performance of the deep learning algorithm ultimately relies on the size and balance of the database. Therefore, for future studies, it is recommended to collect a higher number of point cloud data. This will ensure a larger and more diverse dataset, leading to improved accuracy and generalizability of the weight prediction model.

## 4. Conclusions

Weight prediction of individual free roaming pigs from weight to volume correlation was compared to a PointNet-based 3D deep learning model. The point cloud data of individual free roaming pigs in the holding pen was preprocessed to remove the background scene, and then input into the deep learning architecture for training and testing. The adjusted and unadjusted volumes of the same animals were correlated with scale weight. Despite the training data containing noisy point clouds, the deep learning model outperformed the volume correlation. Tested on unseen point clouds, the model achieved a root mean square error of 6.87 kg with a determination coefficient of 0.9421, whereas the best determination coefficient between volume and weight of pigs was found to be 0.7392. Looking at the different weight classes, both methods performed best for pigs weighing below 55 kg. However, the volume correlation was found to rapidly decline for weight prediction as pigs weight grew, while the deep learning method was found to be stable even for bigger pigs. The results show that 3D deep learning can extract features from the point cloud of pigs in general housing conditions to predict weight, despite adverse conditions with the animal in free movement evidenced by the median correlation between animal weight and volume.

In addition, the findings underscore key aspects for enhancing prediction models through the proposed technique. It is noteworthy that the PointNet architecture underwent training with a limited dataset of 1186 point clouds, each comprised of merely 1500 points on the body surface. Augmenting the training set with a greater number of point clouds is poised to significantly enhance prediction accuracy. The efficacy of the deep learning algorithm is ultimately contingent on the scale and equilibrium of the database. Therefore, for prospective investigations, it is advisable to amass a more extensive dataset of point clouds. This approach ensures a broader and more varied dataset, thereby fostering heightened accuracy and generalization within the weight prediction model.

## Figures and Tables

**Figure 1 animals-14-00031-f001:**
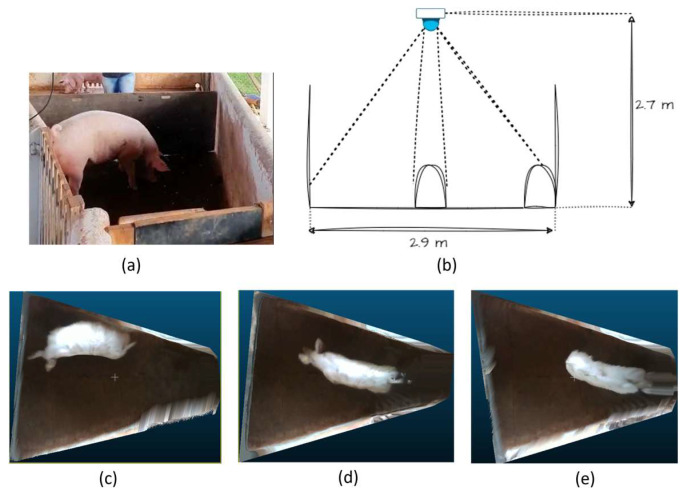
Experimental Setup and collected data; (**a**) pig within the holding pen area while data collection is occurring, (**b**) view of the major axis of data collection holding pen, (**c**–**e**) representative data of pigs in different positions and standing posture within the holding pen area.

**Figure 2 animals-14-00031-f002:**
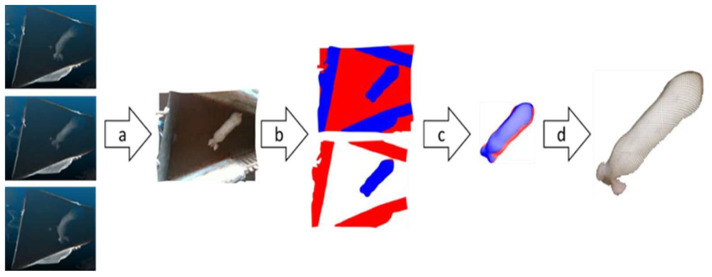
Basic steps for RGB-D image processing to identify and select the animal in the scene (swine): (**a**) image selection; (**b**) thresholding (empty points removal represented by red color); (**c**) color threshold filtering; (**d**) final statistical filtering.

**Figure 3 animals-14-00031-f003:**
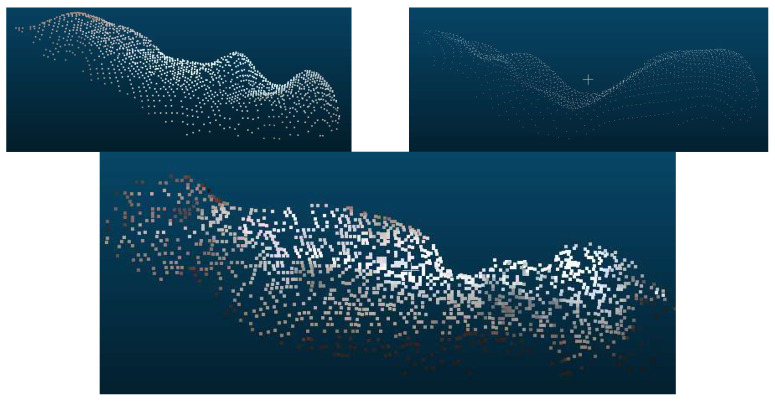
Three representative point cloud data with different quality issues, some unknown wavy pattern, and point clouds with missing large amounts of points.

**Figure 4 animals-14-00031-f004:**
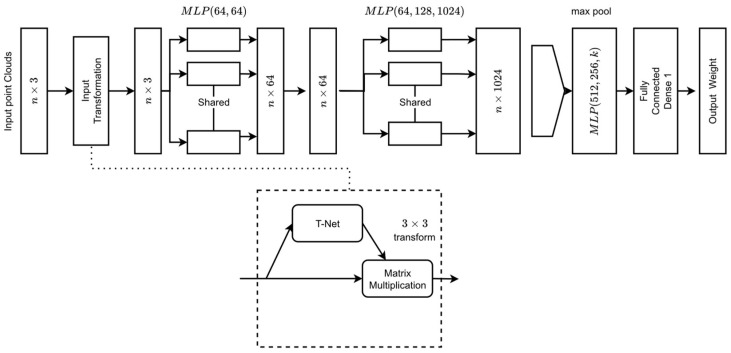
PointNet architecture implemented for weight prediction from a point cloud [15].

**Figure 5 animals-14-00031-f005:**
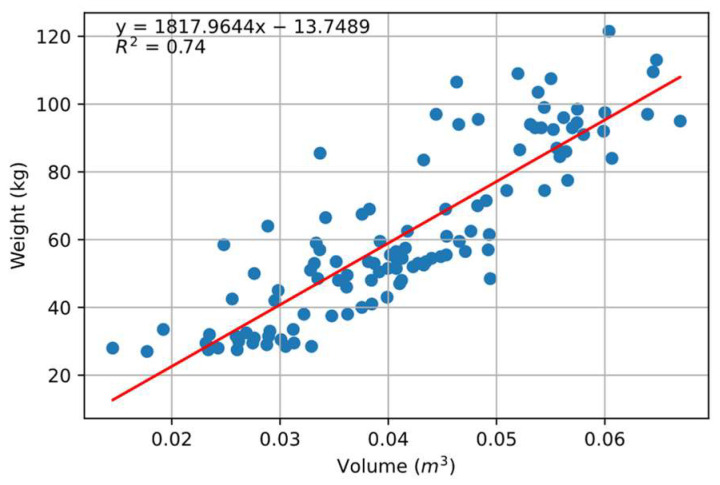
Correlation between the weight of the pig and calculated volume using CloudCompare software’s 2.5D function. Blue dots indicate individual pigs, and the middle red line indicates optimal fitting.

**Figure 6 animals-14-00031-f006:**
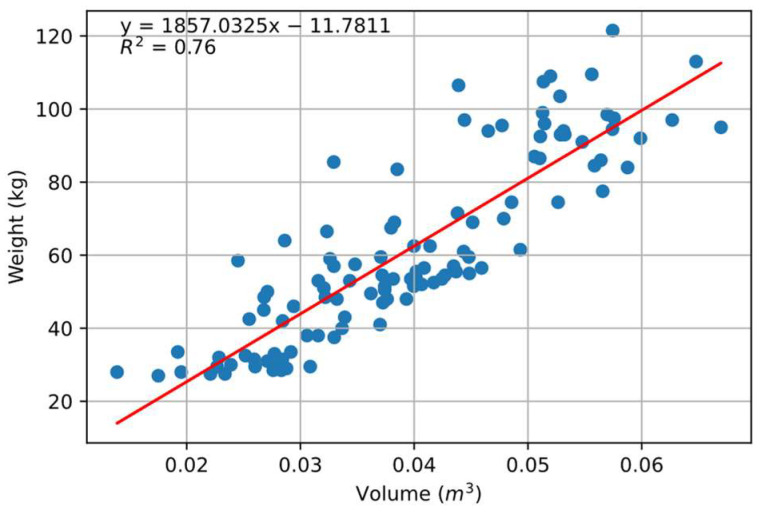
Correlation between adjusted volume as calculated by volume estimation from point cloud after digitally removing head and the neck. Blue dots indicate individual pigs, and the middle red line indicates optimal fitting.

**Figure 7 animals-14-00031-f007:**
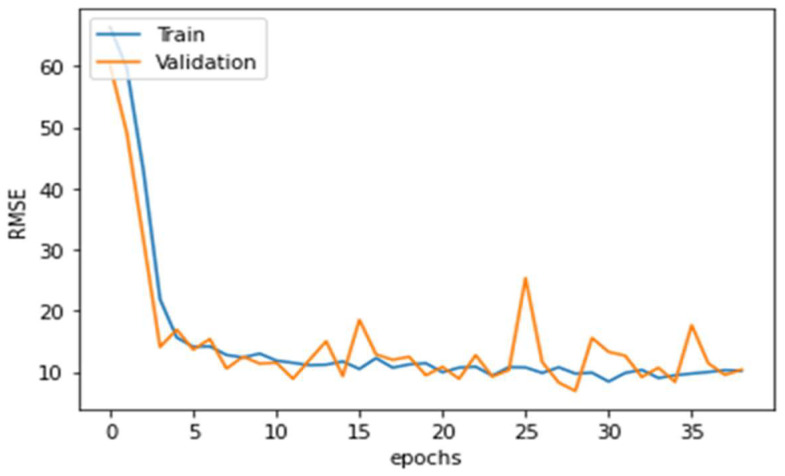
Training history of PointNet architecture over the complete training period.

**Figure 8 animals-14-00031-f008:**
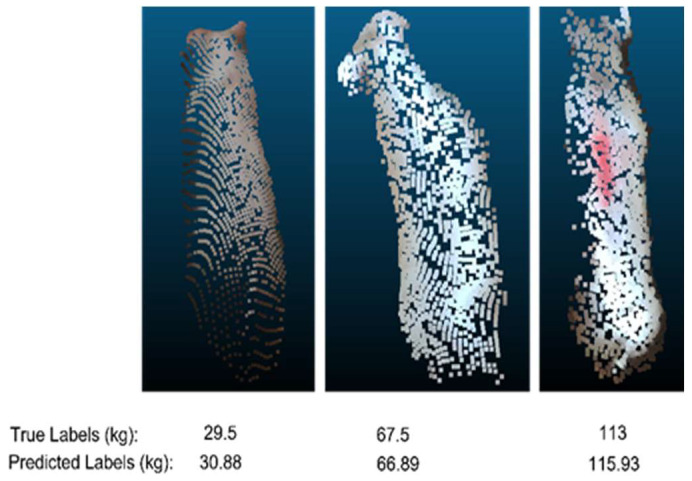
Point clouds, actual and predicted weight values from three different pigs.

**Figure 9 animals-14-00031-f009:**
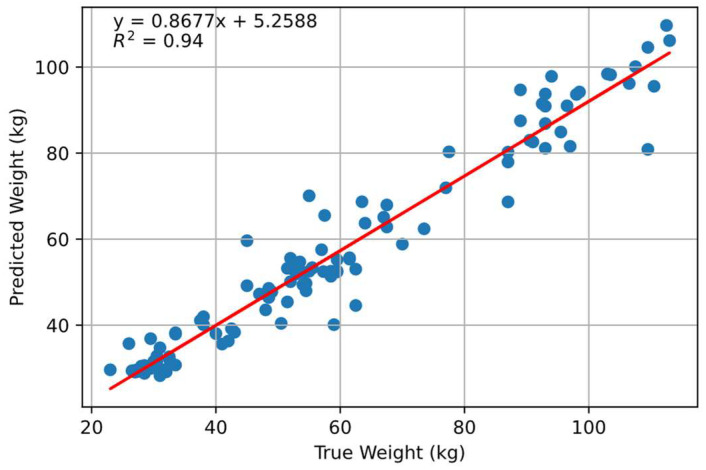
Correlation between predicted weight and scale weight using PointNet model. Blue dots indicate individual pigs, and the middle red line indicates the optimal fitting.

**Table 1 animals-14-00031-t001:** Coefficient of determination and root mean square error (RMSE) of prediction for both the models in each class.

	Volume Correlation		Deep Learning	
Weight Class	Correlation (R^2^)	RMSE (kg)	Correlation (R^2^)	RMSE (kg)
Below 55 kg	0.77	9.16	0.81	4.50
55–90 kg	0.26	13.44	0.0.71	8.20
90–120 kg	0.01	17.03	0.44	9.95
Overall	0.76	12.25	0.94	6.87

## Data Availability

The data presented in this study are available on request from the corresponding author. The data are not publicly available due to ethical reasons.

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
