# Peer review of "Deep Learning Models to Predict Finishing Pig Weight Using Point Clouds"

_animals, 2023, doi:10.3390/ani14010031_

Round 1

Reviewer 1 Report

Comments and Suggestions for Authors

This is a straightforward assessment of two methods to predict finishing pig weights. The methods are sound and the paper is well-written, but with some grammatical and reference errors.

Something to note is that the paper is written in the context that weight prediction of pigs can help with selection of animals to be marketed. However, it seems that the prediction of body weight at marketable weights is very poor. So, is it really that useful for market weight pigs? A discussion on how this data may be useful for the below 55 kg and 55 – 90 kg weight class (where the fit statistics are better) would be useful.

Also, most of the R2 values that are referenced in the text are different than those presented in Table 1. For example, the weight class above 90kg has an R2 value of 0.44 in the text, but 0.66 in the Table. This is a pretty big difference and makes the comparison convoluted.

The in-text reference need to be looked at closely. Some of them don’t have dates, i.e, Brandl et al. (line 57 and 64 among others). Also from looking through other papers in Animals, it looks like the way to cite in-text citations that include the authors name is Okayama et al. [2], not Okayama et al. (2021). Line 131 just has [5].

L19 – add “of” between monitoring and the.

Lines 118 -120: This sentence is confusing and should be reworded. Also “about seven 3D images” in not very precise.

L131: fix reference

L153-155: How did you standardize the cropping out of the head region? With such a large animal, it seems that where you do the cropping could have a pretty significant impact on the volume calculation.

L189: add “the” between from and same

L198: The figure numbers seem out of order. I would suggest moving figure 7 to below this paragraph, renumbering to figure 3 and updating all other figures accordingly.

L222: the is an underscore between weight and estimation.

L251: Does other research support the conclusion that weight and volume are nonlinear?

Comments on the Quality of English Language

The quality of the English good. There are a few grammatical errors, but nothing a solid read through won't fix. 

Reviewer 2 Report

Comments and Suggestions for Authors

This paper describes weight estimation of pigs by RGB-D camera. Although it is stated that it is characterized by its ability to estimate the weight of pigs living in a free environment, there is no description of how this can be achieved in this paper. Just because the environment is not controlled does not prove that it is available in a free environment. No other studies have controlled for brightness, etc. The realsense image is not sensitive to color or brightness in the first place. It describes that the weight could be estimated without shooting from directly above, but it should describe how it could be devised if it were shot from an oblique direction. When photographed from above at an angle, the pigs' bodies are not symmetrical from left to right. This is the reason why many researchers have photographed this from directly above.

As described above, I found no novelty in this paper and judged that it did not meet the criteria for a journal article.

Reviewer 3 Report

Comments and Suggestions for Authors

The authors developed and evaluated a method based on 3D convolutional neural network (PointNet) to predict the weight of pigs from point clouds collected using depth camera. The work is novel and interesting and has potential applications in the food industry. The authors collected the data in a real-world environment and conducted a contrast experiment to demonstrate the advantage of predicting pig weight using 3D CNN.

Possible improvements:

  • Section 2. Materials and Methods should provide a brief summary of the experiments at the beginning, to help readers keep track of the purpose of each subsection.
  • The paper should provide more details on the data collection process, including the height of the camera above the ground, how the recordings were triggered (automated or manual, were there any special conditions), and the camera settings. The paper should also describe the properties of the original data, including the resolution and field of view.
  • Figure 2 should include a reference to PointNet, as the architecture is not designed by this work, and some details in the figure are not explained in the paper (e.g., T-Net).
  • Data augmentation is missing in the experiment. The feature in the point cloud is invariant to translation and rotation, and adding these transformations will greatly reduce the risk of overfitting.

Round 2

Reviewer 1 Report

Comments and Suggestions for Authors

Please take the time to make sure that all the in-text references for the table presented in Table 1 are the same. There are still differences. In the text, you present R2, while in Pearsons. Please standardize. One this is done, please confirm that the values in the table and text line up.

Reviewer 2 Report

Comments and Suggestions for Authors

Eliminating restrictions on the location of pigs and the environment in which they are captured is important and I look forward to further consideration of this issue in the future. Since my questions have been answered appropriately, I will adopt the proposal.
